# Scenarios for a Scaling-up System for Organic Cassava Production in the Mekong River Basin: A Foresight Approach

Benjamas Kumsueb [1], Sukit Rattanasriwong [2], Siviengkhek Phommalath [3], Nareth Nut [4], Jun Fan [5], Hong Xuan Do [6] and Attachai Jintrawet [7,*]

1    Agricultural Resource Management Program, Faculty of Agriculture, Chiang Mai University, Chiang Mai 50200, Thailand; bkumsueb@yahoo.com
2    Independent Researcher, 188/25 Karnkanok Ville 15, Saraphi District, Chiang Mai 50140, Thailand; ssriwong@gmail.com
3    Rice and Cash Crop Research Center, National Agriculture and Forestry Research Institute, Vientiane P.O. Box 7170, Laos; siviengkhek@yahoo.com
4    Faculty of Agricultural Engineering, Royal University of Agriculture, Phnom Penh 12400, Cambodia; nnareth@rua.edu.kh
5    Institute of Food Crops, Yunnan Academy of Agricultural Sciences, Kunming 650200, China; thebestone2007@sina.com
6    Center for Technology Business Incubation and Faculty of Environment and Natural Resources, Nong Lam University—Ho Chi Minh City, Ho Chi Minh City 721400, Vietnam; doxuanhong@hcmuaf.edu.vn
7    Independent Researcher, 133/9 Ban Phai Soi 10, Photharam Road, Chang Phueak, Muang District, Chiang Mai 50300, Thailand
*    Correspondence: attachai.j@cmu.ac.th

**Abstract:** Organic cassava flour and products are in high demand. However, the expansion of organic cassava (OCS) production is rather slow. To increase OCS production, extension workers, cassava flour mills, farmers, and researchers have been collaborating to support the farmers, but the planted areas have remained limited. This research aimed at understanding the current issues in scaling up the organic cassava production. The findings were subsequently used to formulate scenarios and recommendations for the collaborative scale-up of organic cassava production in the Mekong River Basin (MRB). We carried out a six-step foresight process with leaders of organic cassava farmers, the staff of organic cassava flour mills and factories, extension workers, the staff of research agencies, and local policy makers in Thailand. The results revealed two key factors or drivers of changes, namely, the degree of collaboration among stakeholders using multiple-view scenarios or a single-view situation and the degree of learning and communication about OCS that future stakeholders are likely to experience. Four possible scenarios for a scaling-up system of OCS production in the MRB were developed. The foresight process allowed for recognizing multiple views and opinions about the OCS production scaling-up process, considered as a whole system. The system was found to consist of various interdependent components. The process highlighted the need to increase the capacity and opportunities for productive collaboration in research and development. We concluded that the MRB members should issue a policy formulating a joint task force to coordinate the existing institutions' plans and resources towards an actionable OCS production scaling-up system for the MRB in 2030.

**Keywords:** Northeast Thailand; MRB2030; BAU; do it yourself; slow but sure; green dream

## 1. Introduction

Cassava (*Manihot esculenta* Crantz) is widely grown in tropical and subtropical areas of Africa, Asia, and the Americas [1]. In 2022, worldwide, some 32.1 million hectares (Mha) were allocated to cassava production, with an estimated production of fresh storage root or tuber of over 330 million tons. At the global level, the top five producers in terms of cassava planted areas were Nigeria (10.1 Mha), the Democratic Republic of the Congo (6.0 Mha), Thailand (1.6 Mha), Brazil (1.2 Mha), and Côte d'Ivoire (1.1 Mha) [2]. The planted areas and

production of the top five producers accounted for 61.6% and 50.0% of the global cassava planted area and production.

In the Mekong River Basin (MRB) level, a total of 3.40 Mha of cassava planted areas was reported in 2022, which corresponded approximately to 12% of the global cassava planted area. This comprises 3.09 Mha in the MRB member countries Thailand (1.59 Mha), Cambodia (0.76 Mha), Vietnam (0.53 Mha), Lao PDR (0.19 Mha), and Myanmar (0.023 Mha) [2], as well as 0.031 Mha in the Yunnan province of China (PRC) [3]. Statistics for the current state of the OCS production system at the global level are lacking, but organic cassava production management is equally good as that of conventional production practices [4].

Cassava is planted for its edible tubers that have a high carbohydrate content [5]. Cassava leaves are a source of protein, vitamins, and minerals [6]. In the MRB, cassava is predominantly produced by small-holder family farmers, with some external inputs and little knowledge of organic cassava cultivation practices. The natural and gluten-free flour from cassava is witnessing a rise in demand among health-conscious consumers seeking natural and organic products [7]. It is the key unique quality of cassava flour compared to wheat flour. The market for organic cassava (OCS) flour is anticipated to rise at a considerable rate between 2023 and 2030 [8].

Despite the challenges in the transition to organic cassava production and processing [9], Ubon Bioethanol Ltd. Company (UBE) established a system for OCS production in 2018 [10]. In Thailand, the production of OCS flour is based on the collaboration of farmers in managing their fields and of mill operators and owners in maintaining their equipment. Both parties must learn how to transform the internal and external resources at the field and mill levels. At the field level, no chemicals or mineral fertilizers are allowed during the growing season. At the mill level, the processing equipment must be thoroughly cleaned to meet the organic standards of GMP (Good Manufacturing Practice) and HACCP (Hazard Analysis Critical Control Point). The OCS land and flour products must be certified according to the organic cassava standards of the USDA (United States Department of Agriculture), the NOP (National Organic Program), or the European Union Organic system [11].

With an estimated demand of 80,000 tons per year (t/a) of OCS flour for niche markets, 320,000 t/a of fresh, stored OCS tubers must be produced, which requires approximately 14,000 ha/a of OCS cropping area. In 2016, OCS production was started in Ubon Ratchathani province, Northeastern Thailand [11], and was subsequently expanded into Yasothon province in March 2018. The success and sustainability of OCS expanded agricultural production depend on the collaboration of different actors with multiple views and understandings of the situation of small farms and the cassava flour processing mills [12,13].

The World Bank's Training and Visit (T&V) agricultural extension system was introduced in the 1960s [14,15]. The system focuses on the adoption of the green revolution technology to improve agricultural production. It has created subject matter specialists (SMSs) whose role is to formulate recommendations and train and visit farm families. Qamar reviewed agricultural extension systems in developing countries and suggested that reforms must be introduced according to the "situational context" [16]. The "situational context" comprises the prevailing political, institutional, economic, social, cultural, religious, agricultural, geographical, infrastructure, and technological conditions. The Farmer Field School (FFS) system was introduced in the late 1980s and is based on encouraging and stimulating farmers to make their own decisions. The FFS focuses on farmers' capacity to produce transformation based on their situation, supported by relevant research results [17]. A study in Indonesia concluded that the T&V system is appropriate for regions where the level of development is still very low, whereas the opposite is true for the FFS system, which is more successful in developed areas [18]. Agricultural extension systems in MRB member countries share the common goal to improve farmers' livelihoods by applying both systems with some modifications [19–24]. A trusted and successful scaling-up system that accommodates the strengths of various systems and approaches to enhance the capacities of actors is needed and still lacking [25]. An efficient scaling-up system can be

seen as a mechanism to achieve the effective implementation of development plans and programs [26].

With respect to the agricultural extension of the production of OCS, one can expect some success and failures. The success of the transformation can be measured based on its acceptance by local farmers and the consumers of organic products. Consequently, the current extension system may need to be subjected to a major transformation in terms of policy and practices in the MRB, as well as at the national and local levels [27]. The success of the current system is mainly based on the acceptance of a single future for the world development, which is insufficient to handle uncertainties, risks, and opportunities associated with global changes [16]. The multiple futures and objectives approach through the co-formulation of different scenarios is an alternative method to identify the roles and voices of the many concerned individuals and groups to involve [28].

The main research question was what foresight methodology can tell us about the scaling-up system of OCS for MRB2030? The foresight approach is centered around the idea that there is always more than one future, whether possible, preferable, or plausible [29–31]. The foresight methodology is widely used to envisage the future of global agriculture [32,33], digital agriculture in Australia [34], Mediterranean agricultural systems [35], and small farms in Thailand [36]. This is the first study using the foresight methodology to explore the possibilities of OCS production for MRB2030, based on data sets from Thailand.

We used the foresight methodology to investigate and formulate possible future scenarios of OCS production. The purpose of this paper was to present four scenarios for a scaling-up system of OCS production in the MRB for the year 2030—also referred to as "MRB2030". We examined trends and driving factors of changes and formulated scenarios for an OCS scaling-up system (SUS) within MRB2030. The methodology allowed us to understand the studied situation by considering various interacting components and design a holistic approach to upscale the future OCS production.

## 2. Materials and Methods

### 2.1. Study Areas

Our study areas are located in Yasothon and Ubonratchathani provinces in Northeast Thailand, approximately the central part of the lower Mekong River Basin (Figure 1). The organic cassava flour mill (Ubon Bio Ethanol Public Company Limited, Ubon Ratchathani, Thailand) is located in Na Di sub district, Na Yai district, Ubonratchathani Province, and the cassava planting areas are in three districts of Yasothon province. Conventional cassava (CCS) as well as organic cassava (OCS) production methods are employed by some 11,780 farming households in Yasothon province. The cassava planted area in Yasothon province increased rapidly from 8345 ha in 2009 to 14,132 ha in 2020. The Patio district was the first pilot district in the province for OCS production, where a rapid increase in cassava planted area took place in 2020 [37]. The Kham Khuean Kaeo district has the largest area and the highest yield of cassava production in Yasothon province.

### 2.2. Scenario-Based Foresight Process

We followed the six-step foresight process [30] to develop scenarios for the OCS scaling-up system for MRB2030. The six steps are (1) perceiving problematic situations and their context; (2) identifying trends and driving forces based on the STEEP (Society, Technology, Environment, Economy and Policy) model and ranking them by importance and uncertainty; (3) generating an impact/uncertainty matrix; (4) framing the four scenarios; (5) formulating the scenario narratives; (6) developing policies and pathways. The details of each step are described below (with some clarifications available in the Supplementary Materials).

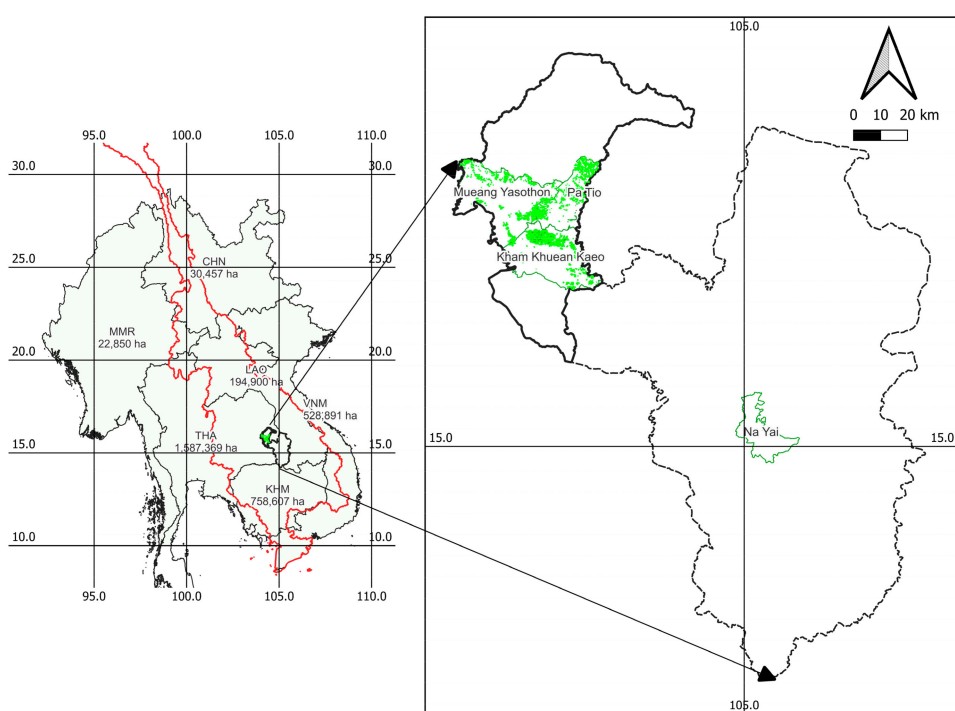

**Figure 1.** Locations of cassava production areas (in green color) in three districts of Yasothon province (solid line boundary), Northeast Thailand and the organic cassava mill in Na Yai district, Ubonratchathani province (dash line boundary), Northeast Thailand. MRB boundaries (in red color) in the inset map (left-hand-side map). Numbers in each country/province boundary indicate cassava planted areas' extension (ha) in 2022.

### 2.2.1. Step 1: Perceiving a Problematic Situation

The main purpose of step 1 was to capture the situation of the OCS system from the literature and key informants. Key informant experts and participants in the project were selected from four groups as follows: (1) fifteen OCS farmers from three districts in Yasothon province; (2) two OCS mill staff members from Ubon Ratchathani province; (3) three staff members of the Department of Agricultural Extension (DoAE), two staff members of the Department of Agriculture (DoA), and two staff members of the National Science and Technology Development Agency (NSTDA); (4) one educator in the Higher Education Institution; (5) one representative of an organic certification organization; (6) nine consumers in Yasothon province. The participants' details are reported in Table S1. During the interview process, the participants were asked to focus on the driving factors [38] of the current method and MRB2030 extension process of OCS production.

### 2.2.2. Step 2: Identifying Trends and Driving Factors

The main aim of Step 2 was to facilitate a discussion about the OCS situation and isolate trends and driving factors, based on the STEEP categories. We also conducted a review of the OCS literature related to the areas [39,40]. In addition, from February to December 2021, the team in Thailand visited OCS farmers in three districts of Yasothon province and talked with them to learn about their concerns and expectations with respect to the upscaling of the system. Subsequently, we organized a horizon-scanning online workshop on 15 January 2022, with participants as shown in Table S1. We combined OCS farmers' concerns and expectations and the experts' opinions regarding the upscaling of the OCS system. We organized, grouped, and validated the trends and driving factors based on the STEEP categories (Table S2).

### 2.2.3. Step 3: Generating an Impact/Uncertainty Matrix

The main aim of Step 3 was to validate and rank the driving factors based on their impact as well as on uncertainties with respect to the upscaling of the OCS system. We organized an online workshop on 7 February 2023 (Table S1). The participants identified the two highest-ranking driving factors with respect to high impact and uncertainty. The two factors were used to create the four scenarios for upscaling the OCS system in MRB2030.

### 2.2.4. Step 4: Framing the Four OCS Scenarios

The main aim of Step 4 was for participants to rank and select trends and driving factors with high impact as well as high uncertainties, to develop axes of change and draft scenarios. Each driving factor was ranked according to its impact and uncertainty as "high", "medium", or "low". The workshop participants agreed and selected two driving factors: collaboration based on scenarios or their own organization's focus (the vertical axis or *y*-axis) and active or passive learning structure, process, and culture (the horizontal or *x*-axis).

### 2.2.5. Step 5: Formulating Scenario Narratives

The main aim of Step 5 was to write up and refine the scenario narratives in four groups. We conducted an online meeting on 7 February 2023 to describe the characteristics and nature of each scenario (Table S2). The group was asked to respond to the question: "What would OCS production in the MRB be like in 2030 in your given scenario?" Each group discussed the scenarios separately. The narratives for future OCS production were based on STEEP and some feedback from the participants. In each scenario, the participants described the OCS production situation in the future of how the situation, actors, and society would change if certain trends and driving forces were to strengthen or diminish, or various events were to occur.

### 2.2.6. Step 6: Developing Policy and Pathways

The main aim of Step 6 was for all participants to formulate policies and pathways towards sustainable OCS production in MRB2030, based on the current situation and the desirable future scenario. The participants of the online meeting on 7 February 2023 provided some opinions for actions towards the "green dream system" as a scenario for the actionable scaling-up system of OCS production in MRB2030.

## 3. Results

The participants and the research teams agreed to develop scenarios for upscaling the OCS production system in the MRB by the year 2030. The MRB is a well-defined geographical area with a standard crop production system adopted by all member countries and mainly operated by small-holder farmers.

### 3.1. Trends and Driving Factors of OCS Production

The scenario development process identified seven trends and drivers of change in the scaling-up system of OCS production in MRB2030, which were classified into five categories according to STEEP (Table 1). The descriptions of each trend and driving factor are presented in Table S2. In Step 2, the participants selected the drivers learning/education capacity and decision making/collaboration as the most important in both the single-view situation and the multiple-view scenarios due to their high impact and uncertainty.

**Table 1.** Trends and uncertainty of OCS production according to the STEEP categories.

| Category | Trend/Driving Factor | Impact | Uncertainty |
|---|---|---|---|
| Social | TS1: Aging population with low education, learning unrelated to the opportunities of OCS | High | High |
| Technology | TT1: Easy access to agrochemicals | High | Low |
| | TT2: Information and communication technology improving communication and efficiency | Low | Low |
| Environment | TEn1: Declining qualities and limited quantities of land resources | High | Low |
| Economic | TEc1: Increasing inequity | Low | Low |
| | TEc2: Inside-village and outside-village income opportunities | Low | Low |
| Policy | TP1: Centralized decision making, with short-term gain | High | High |

*3.2. Driving Factors with High Impact and High Uncertainty*

During the final meeting, the participants agreed to choose two driving factors, based on two criteria, which were (1) the significance and importance of their impact; (2) the degree of uncertainty in the upscaling of OCS production in MRB2030. Subsequently, the two driving factors were identified as (1) the practices of collaboration based on the single-view or multiple-view scenarios; (2) the capacity and opportunities to deliver, learn, and communicate about OCS.

3.2.1. OCS Policy and Practice Based on Single-View or Multiple-View Scenarios with Respect to OCS Expansion

The vertical axis relates to the extent of policy uncertainty and change, for example, decisions based on single-view or multiple-view scenarios, regarding, e.g., subsidies for chemical agriculture and the associated consequences. A single-view scenario policy that supports chemical use in agriculture can impact environmental issues in the long term and increase uncertainty about upscaling the OCS value chain and production.

3.2.2. Capacity and Opportunities to Deliver, Learn. and Communicate about Organic Products and Markets, Especially Related to Cassava

The horizontal axis relates to the level of communication capacity of the whole society regarding organic cassava. With an increased communication capacity, collaboration at all levels will provide a better understanding of the benefits of the OCS transformation. However, uncertainty related to international markets is rather complex, and competition may generate ups and downs, which will ultimately influence the farm gate price of raw organic cassava.

*3.3. The Four Scenarios*

During the final meeting, the participants agreed to choose two driving factors, based on the significance of their impact; the degree of uncertainty; and their importance for the upscaling system of OCS production in the MRB. The two selected driving factors were "collaboration based on the single-view or multiple-view scenarios" and "the capacity and opportunities to learn and communicate about OCS". Table 2 presents the descriptions of the four scenarios. Figure 2 provides an illustration of the four scenarios of the OCS scaling-up system.

**Table 2.** Matrix of driving factors and characteristics of the four scenarios for the OCS scaling-up system in MRB2030.

| Driving Factor | BAU (Business as Usual) | Do It Yourself | Slow but Sure | Green Dream System |
|---|---|---|---|---|
| Driving factor | | | | |
| Collaboration and decision making based on the single-view scenario | Single | Single | Multiple | Multiple |
| Capability to communicate and learn | Low | High | Low | High |
| Characteristics of each scenario for MRB2030 based on STEEP categories | | | | |
| Social | No visible structure, process and culture operated by OCS farmers. The upscaling of the system is operated by the central government's extension agency. No partnerships of OCS suppliers and mills. Focus on learning about technology | Top-down approach for OCS farmers' organizations scattered in MRB2030. The upscaling of the system is operated by the central government's extension agency with some weak organization and limited partnership with the OCS mill and suppliers. Focus on learning to adopt technology | Bottom-up approach will establish OCS farmer organizations in selected areas in the MRB. Weak partnership with OCS mill and suppliers. Focus on learning about technology. | Well-organized structure, process, and culture organized and operated by OCS farmers. Trusted partnerships of OCS suppliers and mills. Focus on human resources with skill in social learning and social/individual entrepreneurship |
| Technology | CCS-based from silo SRI/SREL system, education, training, no career path system | Mixed CCS and OCS practices. Some investment in an OCS technology development system | Mixed CCS and OCS practices, technology research, development, and system implementation | Inclusive OCS SREL system with education, learning to ask questions, collaborate. and co-invest in OCS technology research, development, and system implementation |
| Environment | CCS affecting the environment, high GHG emission. | Medium GHG emission. | High GHG emission | Low GHG emission based on science and data systems |
| Economy | OCS as a part-time source of income, limited economic opportunities and investment | OCS as a major source of income for some farmers | OCS as a part-time activity and minor source of income, limited economic opportunities | OCS as the major source of income and investment |
| Policy | Top-down policy, linked with profit-oriented financial institutions and mill operators. No OCS SRI/SREL investment for the improvement of productivity and quality | Top-down, one-way linked with financial institutions, farmers, mill operators, and consumers. Some SRI investment for upscaling of OCS production, improving productivity, and applying some techniques to improve product quality | Transformation to OCS production takes time due to low capabilities to communicate and learn | Supports the collaboration between financial institutions, farmers, organic cassava mill operators, and consumers. Co-investment on SRI/SREL to improve OCS quality and productivity and support systemic upscaling, based on science and data |

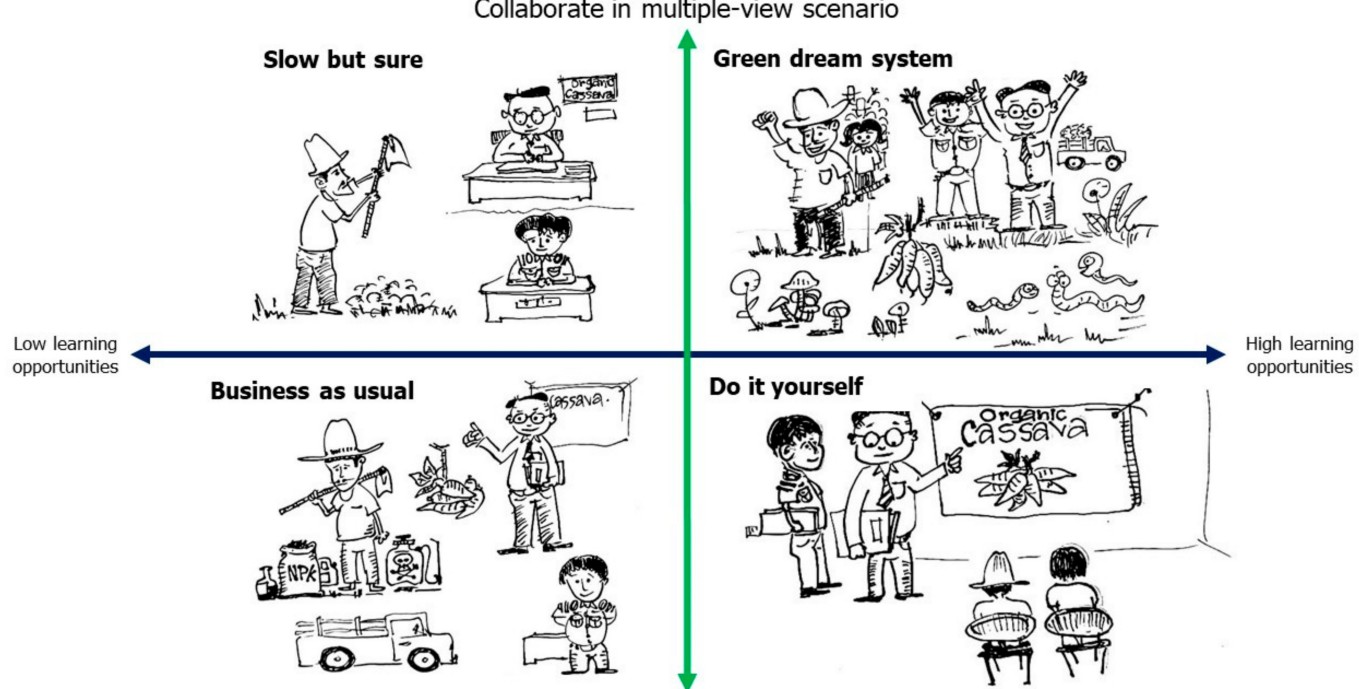

**Figure 2.** The four scenarios for the scaling up of OCS production in MRB2030, based on a data set regarding Yasothon province, Northeast Thailand.

### 3.3.1. The Business as Usual (BAU) Scenario

In 2030, the extension system based on conventional agriculture with the use of agrochemicals will be the only learning system in the MRB. The learning capacity about OCS will still be low, and collaboration will still be that of the single-view or silo business model. The capacity to share new knowledge will be insufficient to shift to the upscaling model. Meanwhile, the demand for organic products from cassava will have decreased at the global level. The science, research, education, and learning (SREL) system will be unable to provide OCS solutions for farmers to benefit from opportunities and cope with uncertainties. This will be mainly due to ineffective and inefficient collaboration, learning, and communication of various agencies. The system operates based on a red-tape and bureaucratic, technocrat, and highly centralized decision-making business model. The SREL system will remain a closed system. The organic certification system will remain out of reach to most small-holder OCS farmers. There is no career path that could attract the young generation into the low-learning and single-view collaboration model. The financial system will be stuck in the one-way short-term investment model, i.e., loan to farmers for the bank's profits. Subsequently, this scenario will lead to a linear learning and collaboration model for the chemical-based production and processing of cassava in the MRB.

### 3.3.2. The Do It Yourself Scenario

The participants agreed that this scenario is a step forward and a likely mode of operation towards the 2030 green dream system. The system upscaling will allow selected groups of farmers with high learning capability to collaborate using a range of information and communication technologies for decision making, i.e., AI (artificial intelligence). In some areas, the learning capabilities with respect to OCS will dramatically shift towards the new mode of knowledge-driven OCS production. However, the extension system based on conventional cassava production methods will be the dominant learning system in the MRB. Collaboration will remain similar to that in the single-view, individualistic business model. The OCS mill and public agencies including SREL institutions will be able to

collaborate, establish partnerships, and offer some solutions. The SREL system will remain a closed system. The solutions will be based on one's agenda and priorities. The majority of small-holder cassava farmers, both conventional and organic groups, will be unable to establish organizations and create opportunities to shift to knowledge sharing and learning. Some selected farmers will be able to benefit from knowledge-sharing opportunities, with some support from the SREL system. The MRB will present various successful cases of OCS production for niche markets but will remain unable to scale it up.

### 3.3.3. The Slow but Sure Scenario

In 2030, due to the low level of learning capabilities, the conventional cassava extension system will be dominant in spite of a certain upscaling of the system in the MRB. The collaboration among stakeholders will be mostly concentrated on conventional cassava production and processing. The production and processing of cassava in the MRB will be trapped in a system relying on the use of agro-chemicals. OCS mills and public agencies, including SREL institutions, will be able to collaborate, establish partnerships, and offer some solutions, but unable to scale up the OCS production due to the low learning capabilities. Also, small-holder cassava farmers will cooperate and establish local organizations to motivate the young generation to follow an OCS production career path. The MRB will present various successful cases of OCS production but, again, will be able to scale it up rather slowly.

### 3.3.4. The Green Dream System Scenario

In this scenario, the common goal is that methods relying on agro-chemicals for cassava production and processing will be abandoned by 2030. This will be mainly due to high learning and communication capabilities and collaboration among stakeholders. However, some conventional cassava production in some districts will continue to exist to provide local markets. The MRB member countries will have collaboratively established an "organic agricultural corridor", with incentives to attract investments and talented human resources. OCS processing in the MRB will have undergone a revolution, fully accepted by society and policy. The consumption of organic products from cassava in 2030 will have increased both in the MRB and on the global market, due to the awareness of climate change and sustainable farming practices, as well as to the rising incomes in this context of economic expansion. The range of organic products will be greater than that predicted in MRB2030. They will provide added value and more consumers options under some partnership systems with producer networks. Small farms will remain the key producers and supply the raw material to processing mills all year round. Farmers will have a high learning capability to collaborate with the other stakeholders using a range of information and communication technologies, i.e., AI. They will be free in the decision to shift from conventional to organic cassava production. The new partnerships along with the new structure, process, and culture will shift towards knowledge-driven OCS production. Curriculum improvement and skill training in collaboration are the central focus of human resource development and professionalism for organic cassava production. Organic cassava farming is a professional career path well respected by the society. The newly opened SREL system will be established and staffed with highly qualified professionals. Economic rewards will be linked to the well-being of OCS communities in the respective areas. The financial system will provide long-term investments which offer solutions based on the SREL system. This will lead to a full circularity of the production and processing of OCS in the MRB.

## 4. Discussion

The presented scenarios are representations of an effort to think and scan across a range of future possibilities, not predictions of the future, for the upscaling of the OCS production system [28,38,41]. The findings and their implications should be discussed in the broadest possible context. Future research directions will also be highlighted. However,

the participants agreed that it is unlikely that the MRB region will follow the exact path of any of the four scenarios for the upscaling of the system, but MRB2030 will likely exhibit some aspects of all four scenarios. The scenario development process helped identify driving factors of the pathway upon which the upscaling of the OCS system in the MRB is developing. The BAU scenario can be used as a reference, representing the current upscaling system, to co-formulate various pathways and policies towards the green dream scenario that can be developed and implemented. The two driving factors were policy and practices based on the single-view or multiple-view scenarios and the capacity to deliver, learn, and communicate for a better understanding of organic cassava production and processing through education and learning systems.

### 4.1. Learning and Education for an OCS Scaling-up System in the MRB

Learning and education are the key factors for human resource development. MRB2030 green dream scenario needs a new policy for actors in school and higher education institutions in the MRB to collaborate and redesign the current curriculum and training contents and materials to educate young professionals with knowledge of OCS production and related skills, ready to follow this career path [42]. This policy must be shifted towards changes in structure, procedure, and culture of learning and education to transform the status quo into a desirable future [43]. The policy must couple formal and informal education and training for job creation and an industry built around the production and processing of organic cassava products [44,45]. Actors in the MRB region need a roadmap to develop and enhance the capacity of human resources in the essential field of the economy, in particular, the upscaling of the OCS system [46,47]. The roadmap at the learner level is key to enhancing the capacity to collaborate in order to benefit from new opportunities. One must gain a better understanding of sustainable organic cassava production and scientific knowledge, especially with long-term field experiments. This understanding will create awareness that the yield of organic production in developed countries is 40% of that of conventional production. Thus 67% more of agricultural land is needed to produce the same amount of crop [41,48]. However, high productivity and yield stability over time have been achieved in organic agriculture through the maintenance of soil fertility and functions of the soil with agronomic practices such as balanced crop rotations and the application of organic amendments [49]. Therefore, the learning space or "learning ecosystem" in the MRB must be redesigned to incorporate the numerous existing organic cassava farms and organic cassava processing mills. This is an essential learning ecosystem for actors along the whole value chain of the upscaled OCS production system. This new work force of the upscaled system will be a key agent for change towards a full-scale OCS production in the MRB. Finally, the rewarding system must also be adjusted to accommodate the potential of the new work force [50,51].

### 4.2. Collaboration in OCS System Upscaling in the MRB

The collaboration of actors will lead to a higher capability to handle risks associated with the new structures, procedures, and culture practices of the OCS upscaling system. The collaboration process also needs time, care, resources, data sets, monitoring, verification, and reporting of the project's progress [52]. The existing public-funded programs and projects can be refocused from a short-term and techno-centric to a long-term and human-centric collaboration platform [53]. The platform will promote the exchange of knowledge and understanding of the nature and dynamics of organic cassava production and processing. Such understanding will support the capacity to predict the consequences of decision making to allocate resources for the efficient OCS production in the MRB towards the green dream scenario.

### 4.3. Human Resources for an OCS Scaling-up System in the MRB

Human resources are a key driver at all levels of system upscaling [26,54]. At the local institutional level, the leaders of farmers' groups can lead discussions about self-

motivation for the OCS transformation. The leaders can establish groups of pioneers in villages to learn and communicate the pros and cons of OCS production. An OCS group or cooperative can be established at the village level to provide various materials for widespread learning. The learning process should improve the capacity for organic cassava production, monitoring, reporting, and verification of the learners. The newly formed cooperatives may jointly organize series of learning sessions about OCS held by researchers and other external experts. In addition, the cooperatives can also provide supplies of materials for OCS production, i.e., organic fertilizers and integrated pest management materials, to the newcomers.

At the national level, the leaders of science, research, innovation, and extension agencies can collaborate to establish new funding for a research program that focuses on upscaling OCS system. The program must be based on the diversity, autonomy, openness, non-ideological character, and collaboration of the OCS cooperatives, researchers, and mills to develop solutions that will have a positive impact [55]. The program should include OCS scholarships for multi-generational learning and implementation.

At the vocational and higher education (VHE) institution level in agriculture, the leaders of VHE must refocus their resources from a chemical-based to an organic-based education, learning, and knowledge creation platform [56]. With this new platform, OCS-based materials can be co-created, shared, and co-learned by learners in classrooms and in the organic cassava production plots and farmers. This type of co-creation green learning has been discussed and successfully implemented in green analytical chemistry [57], green jobs [58], and sustainable agriculture and natural resource management [59].

### 4.4. Policy Implications of Our Study Relevant to the OCS Scaling-up System in the MRB

To achieve the desirable scenario for the sustainable upscaling of OCS production within MRB2030, the participants also suggested that three components must be in place. Firstly, the MRB has to have the capacity to improve the interaction between education, collaboration, small and medium enterprise (SME) entrepreneurship, and the financial systems. Secondly, a new approach is needed to support a network of leaders and actors in system upscaling at the local and the MRB levels. Thirdly, political will among the MRB members to establish a task force or a consortium to jointly plan, implement, check, and work for an actionable scaling-up system of OCS production in the MRB is necessary [60].

The foresight methodology provided four scenarios based on two driving factors, namely, collaboration in the single-view or multiple-view scenarios and the learning opportunity. However, we need to collect and integrate data sets and opinions on system upscaling from organic production and supply chains in Cambodia, China, Lao, Myanmar, and Vietnam. The support of joint task forces of MRB members is required. These joint task forces can localize areas where to implement the foresight process, conducting it in the local languages of the MRB region. This will allow for a better understanding, based on data sets and opinions, of the trends and driving factors of the necessary changes toward a scaling-up system of OCS production.

### 4.5. Limitations of the Study

Our current effort has some limitations that point to the importance of future collaborative research as follows:

(1) The foresight process depends on the quantity and quality of the available data [61]. We need to co-formulate an innovative way to build trust in the participants from the MRB member countries and encourage them to express their opinions regarding each local situation, which is a challenge in Thailand [62] and Southeast Asia cultures [63]. The field visit by the first author was a practical approach to engage the farmers and leaders in the field to frankly express their opinions. Future research efforts must recognize the diverse cultural and contextual aspects of the MRB and systemically learn and formulate actionable strategies to deal with them.

(2)    From the methodological point of view, the identification of problematic situations heavily relies on the perspectives of key informants and experts. This approach may introduce bias and subjectivity, potentially overlooking certain factors or perspectives that could be crucial for understanding a situation comprehensively. In addition, the selection of key informants and participants for interviews and workshops should be opened to include consumers, environmental organizations, and international agencies. Finally, the four scenarios should be tested to accurately demonstrate the complexity of potential future developments of OCS production in MRB2030.

## 5. Conclusions

In this article, the foresight method was used to bring participants with diverse backgrounds and experiences in OCS production together to brainstorm and visualize a system upscaling within MRB2030. The participants expressed their concerns and opinions on the situation by considering various interacting components. The findings indicated trends in five categories, namely, Society, Technology, Environment, Economic, and Policy, i.e., STEEP. The significant finding of this study was the identification of two driving factors that would influence the OCS scaling-up system in MRB2030. The process also yielded four possible scenarios for the MRB2030 OCS upscaling system: the business as usual, the do it yourself, the slow but sure, and the green dream system. The efficient scaling-up system of organic cassava production can be considered as opportunity for MRB members to achieve the common goal of extended OCS production in 2030. The major implication of our research is a need for MRB members to establish a joint task force for the necessary changes toward the scaling-up system of OCS production, a production system that depends on both internal and external resources.

**Supplementary Materials:** The following supporting information can be downloaded at: https://www.mdpi.com/article/10.3390/agriculture14040600/s1, Table S1: The participants in the OCS production project for MRB2030; Table S2: Trends and driving factors of OCS production towards MRB2030.

**Author Contributions:** B.K., S.R., S.P., N.N., J.F., H.X.D. and A.J. contributed equally to the study design, data collection, data analysis, data interpretation, writing and editing the manuscript draft. B.K., S.R. and A.J., supervision. All authors have read and agreed to the published version of the manuscript.

**Funding:** This research was funded by the National Research Council of Thailand (NRCT), Thailand, grant number PHD/0231/2560.

**Institutional Review Board Statement:** Not applicable.

**Data Availability Statement:** Data are contained within the article and Supplementary Materials.

**Acknowledgments:** We appreciate editors and anonymous reviewers for their valuable suggestions and comments. We appreciate Roungchai Ketsakorn for the cartoon representation of the four scenarios in Figure 2.

**Conflicts of Interest:** The authors declare no conflicts of interest.

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
