# Peer review of "Scenarios for a Scaling-up System for Organic Cassava Production in the Mekong River Basin: A Foresight Approach"

_agriculture, doi:10.3390/agriculture14040600_

Round 1

Reviewer 1 Report

Comments and Suggestions for Authors

Dear authors,

The paper is very interesting, but it should be revised according to the following suggestions:

The introduction provided effectively outlines the current status and significance of cassava production, particularly within the Mekong River Basin (MRB), while also addressing the emerging trend of organic cassava production. By citing specific statistics regarding cassava production worldwide and within the MRB, the introduction establishes a clear context for the subsequent discussion. The lack of explicitly stated research questions or hypotheses in this introduction is notable. While the introduction sets the stage for discussing the challenges and opportunities associated with organic cassava production, it could benefit from explicitly stating the research objectives or questions that the study aims to address. This would provide readers with a clearer understanding of the study's focus and direction.

However, the introduction does effectively identify the gap in the literature by highlighting the need for a scaling-up system for organic cassava production within the MRB. It discusses the limitations of the current agricultural extension systems and emphasizes the necessity for a more comprehensive approach to address the challenges of scaling up organic cassava production. Overall, while the introduction provides valuable context and identifies the research gap, explicitly stating research questions or hypotheses would enhance its clarity and focus, thereby guiding readers more effectively through the subsequent discussion.

While the methodology outlined presents a structured approach to developing scenarios for the scaling up of organic cassava production within the Mekong River Basin (MRB) for the year 2030, there are several weak points that should be addressed:

1. Subjectivity in perceiving problematic situations: the identification of problematic situations heavily relies on the perspectives of key informants and experts. This approach may introduce bias and subjectivity, potentially overlooking certain factors or perspectives that could be crucial for understanding the situation comprehensively.

2. Limited representation of stakeholders: the selection of key informants and participants for interviews and workshops seems to focus mainly on specific groups, such as OCS farmers, mill staff, and government officials. This limited representation may lead to an incomplete understanding of the broader stakeholder landscape, potentially neglecting the perspectives of other relevant actors, such as consumers, environmental organizations, or international agencies.

3. Lack of transparency in ranking and selection process: the methodology mentions the ranking and selection of driving factors and scenarios by workshop participants, but it lacks transparency regarding the criteria used for ranking and selection. Without clear criteria or guidelines, the process may be susceptible to biases or inconsistencies.

4. Limited validation of scenarios: while the methodology describes the formulation of scenarios based on identified driving factors, there is limited discussion on how these scenarios were validated or tested for robustness. Without validation, the scenarios may lack credibility or fail to capture the complexity of potential future developments accurately.

5. Insufficient discussion on policy implications: although the methodology includes a step for developing policy and pathways, there is limited detail provided on the specific policy recommendations or pathways identified by participants. A more comprehensive discussion on the policy implications of each scenario and the feasibility of proposed actions would enhance the practical relevance of the study's findings.

Addressing these weak points could strengthen the rigor, transparency, and relevance of the scenario-based foresight process for scaling up organic cassava production in the MRB.

The results should be improved according to the following suggestions:

While the study outlines the scenario development process and identifies key driving factors, there are instances where the information is repeated or presented in a slightly disjointed manner. Ensuring clarity and consistency in the presentation of information would enhance the readability and understanding of the results. The study identifies driving factors such as collaboration based on single view or multi-views/scenarios and the capacity and opportunity to learn and communicate about organic cassava systems. However, the rationale behind selecting these driving factors could be elaborated further. Providing more detailed explanations or citing specific evidence to justify the selection of these factors would strengthen the validity of the findings. Conducting sensitivity analysis or robustness checks could help assess the stability and reliability of the scenario outcomes. By varying key assumptions or parameters and examining how the results change, researchers can better understand the range of possible future trajectories and identify potential areas of uncertainty or risk. 

Overall, the discussion section provides a comprehensive analysis of the findings and their implications for scaling up organic cassava production within the Mekong River Basin (MRB) by 2030. However, there are a few points that could be improved:

E.g. integration of stakeholder perspectives: the discussion touches on the importance of collaboration and education for achieving sustainable upscaling of organic cassava production, but it could delve deeper into the integration of diverse stakeholder perspectives in the decision-making process. Incorporating insights from a broader range of stakeholders, including farmers, policymakers, and consumers, could enrich the discussion and lead to more inclusive and effective strategies.

or

Consideration of cultural and contextual factors: the discussion acknowledges challenges in engaging stakeholders and building trust, particularly in the context of Southeast Asian cultures. Further exploration of these cultural and contextual factors, along with strategies for overcoming them, would provide valuable insights for researchers conducting similar studies in diverse cultural settings.

The conclusion does not contextualize the significance of the study within the broader field of agricultural foresight or sustainable development. Providing a brief contextualization could help readers understand the broader implications of the research.

Author Response

We thank the Reviewer for their thoughtful comments. Below please find comments from the Reviewer (in black) and our responses (in blue).

Reviewer 2 Report

Comments and Suggestions for Authors

Scenarios of Scaling up System for Organic Cassava Production in Mekong River Basin: A Foresight Approach

From the introduction, the authors need to include the global total cassava production in 2022. Which country is the highest producer of cassava globally? What is the rank of Thailand in the global production of cassava? In order to present a robust background information on cassava, especially a global perspective, authors are expected to present these statistics from reputable organizations like FAO statistical database (FAOSTAT) 2024.

From my own view, authors have not been able to describe the concept of "foresight approach" employed in this study. This make it difficult to clearly identify the knowledge gap(s) that this study sets to fill in literature.

The current state of OCS production system globally is conspicuously missing in this study. Authors are expected to commence the study with global view on OCS production system before focusing on the MRB using foresight approach.

Kindly give full meaning of STEEP at first mention. Also, your sampling procedure is quite unclear to me. From figure 2, why is the labelling of the diagrams starting from 3. It should start from 1-4 and not from 3-2.

Before the conclusion section, kindly present a sub-section with the heading "limitations of the study"

Comments on the Quality of English Language

Fine

Author Response

We thank the Reviewer for thoughtful comments. Below please find comments from the Reviewer (in black) and our responses (in blue).

Round 2

Reviewer 1 Report

Comments and Suggestions for Authors

The paper was significantly improved according to reviewer comemnts and suggestions.

Author Response

We appreciate the valuable comments from Reviewer 2 and we have added our responses in the attached file.

Reviewer 2 Report

Comments and Suggestions for Authors

Many thanks for sending your revised manuscript. I would like the authors to add the production levels of top 5 cassava producers in 2021 or 2022.

However, after this inclusion, the editor can make final publication decision on the revised manuscript. Thank you.

Comments on the Quality of English Language

Fine

Author Response

We appreciate the comment and we have added cassava planted areas and production in Line 46-50 (in blue in the revised version).
